# Pruning neural networks without any data by iteratively conserving synaptic flow

**Hidenori Tanaka**[*]
Physics & Informatics Laboratories
NTT Research, Inc.
Department of Applied Physics
Stanford University

**Daniel Kunin**[*]
Institute for Computational and
Mathematical Engineering
Stanford University

**Daniel L. K. Yamins**
Department of Psychology
Department of Computer Science
Stanford University

**Surya Ganguli**
Department of Applied Physics
Stanford University

## Abstract

Pruning the parameters of deep neural networks has generated intense interest due to potential savings in time, memory and energy both during training and at test time. Recent works have identified, through an expensive sequence of training and pruning cycles, the existence of winning lottery tickets or sparse trainable subnetworks at initialization. This raises a foundational question: can we identify highly sparse trainable subnetworks at initialization, without ever training, or indeed *without ever looking at the data*? We provide an affirmative answer to this question through theory driven algorithm design. We first mathematically formulate and experimentally verify a conservation law that explains why existing gradient-based pruning algorithms at initialization suffer from layer-collapse, the premature pruning of an entire layer rendering a network untrainable. This theory also elucidates how layer-collapse can be entirely avoided, motivating a novel pruning algorithm *Iterative Synaptic Flow Pruning (SynFlow)*. This algorithm can be interpreted as preserving the total flow of synaptic strengths through the network at initialization subject to a sparsity constraint. Notably, this algorithm makes no reference to the training data and consistently competes with or outperforms existing state-of-the-art pruning algorithms at initialization over a range of models (VGG and ResNet), datasets (CIFAR-10/100 and Tiny ImageNet), and sparsity constraints (up to 99.99 percent). Thus our data-agnostic pruning algorithm challenges the existing paradigm that, at initialization, data must be used to quantify which synapses are important.

## 1 Introduction

Network pruning, or the compression of neural networks by removing parameters, has been an important subject both for reasons of practical deployment [1, 2, 3, 4, 5, 6, 7] and for theoretical understanding of artificial [8] and biological [9] neural networks. Conventionally, pruning algorithms have focused on compressing pre-trained models [1, 2, 3, 5, 6]. However, recent works [10, 11] have identified through iterative training and pruning cycles (*iterative magnitude pruning*) that there exist sparse subnetworks (*winning tickets*) in randomly-initialized neural networks that, when trained in

---

[*]Equal contribution. Correspondence to `hidenori.tanaka@ntt-research.com` and `kunin@stanford.edu`.

isolation, can match the test accuracy of the original network. Moreover, its been shown that some of these winning ticket subnetworks can generalize across datasets and optimizers [12]. While these results suggest training can be made more efficient by identifying winning ticket subnetworks at initialization, they do not provide efficient algorithms to find them. Typically, it requires significantly more computational costs to identify winning tickets through iterative training and pruning cycles than simply training the original network from scratch [10, 11]. Thus, the fundamental unanswered question is: can we identify highly sparse trainable subnetworks at initialization, without ever training, or indeed *without ever looking at the data*? Towards this goal, we start by investigating the limitations of existing pruning algorithms at initialization [13, 14], determine simple strategies for avoiding these limitations, and provide a novel data-agnostic algorithm that achieves state-of-the-art results. Our main contributions are:

1. We study *layer-collapse*, the premature pruning of an entire layer making a network untrainable, and formulate the axiom *Maximal Critical Compression* that posits a pruning algorithm should avoid layer-collapse whenever possible (Sec. 3).
2. We demonstrate theoretically and empirically that *synaptic saliency*, a general class of gradient-based scores for pruning, is conserved at every hidden unit and layer of a neural network (Sec. 4).
3. We show that these *conservation laws* imply parameters in large layers receive lower scores than parameters in small layers, which elucidates why single-shot pruning disproportionately prunes the largest layer leading to layer-collapse (Sec. 4).
4. We hypothesize that *iterative magnitude pruning* [10] avoids layer-collapse because gradient descent effectively encourages the magnitude scores to observe a conservation law, which combined with iteration results in the relative scores for the largest layers increasing during pruning (Sec. 5).
5. We prove that a pruning algorithm avoids layer-collapse entirely and satisfies Maximal Critical Compression if it uses iterative, positive synaptic saliency scores (Sec. 6).
6. We introduce a new data-agnostic algorithm *Iterative Synaptic Flow Pruning (SynFlow)* that satisfies Maximal Critical Compression (Sec. 6) and demonstrate empirically[2] that this algorithm achieves state-of-the-art pruning performance on 12 distinct combinations of models and datasets (Sec. 7).

## 2 Related work

While there are a variety of approaches to compressing neural networks, such as novel design of micro-architectures [15, 16, 17], dimensionality reduction of network parameters [18, 19], and training of dynamic sparse networks [20, 21, 22], in this work we will focus on neural network pruning.

**Pruning after training.** Conventional pruning algorithms assign scores to parameters in neural networks *after* training and remove the parameters with the lowest scores [5, 23, 24]. Popular scoring metrics include weight magnitudes [4, 6], its generalization to multi-layers [25], first- [1, 26, 27, 28] and second-order [2, 3, 28] Taylor coefficients of the training loss with respect to the parameters, and more sophisticated variants [29, 30, 31]. While these pruning algorithms can indeed compress neural networks at test time, there is no reduction in the cost of training.

**Pruning before Training.** Recent works demonstrated that randomly initialized neural networks can be pruned *before* training with little or no loss in the final test accuracy [10, 13, 32]. In particular, the Iterative Magnitude Pruning (IMP) algorithm [10, 11] repeats multiple cycles of training, pruning, and weight rewinding to identify extremely sparse neural networks at initialization that can be trained to match the test accuracy of the original network. While IMP is powerful, it requires multiple cycles of expensive training and pruning with very specific sets of hyperparameters. Avoiding these difficulties, a different approach uses the gradients of the training loss at initialization to prune the network in a single-shot [13, 14]. While these single-shot pruning algorithms at initialization are much more efficient, and work as well as IMP at moderate levels of sparsity, they suffer from layer-collapse, or the premature pruning of an entire layer rendering a network untrainable [33, 34]. Understanding and circumventing this layer-collapse issue is the fundamental motivation for our study.

# 3 Layer-collapse: the key obstacle to pruning at initialization

Broadly speaking, a pruning algorithm at initialization is defined by two steps. The first step scores the parameters of a network according to some metric and the second step masks the parameters (removes or keeps the parameter) according to their scores. The pruning algorithms we consider will always mask the parameters by simply removing the parameters with the smallest scores. This ranking process can be applied globally across the network, or layer-wise. Empirically, its been shown that global-masking performs far better than layer-masking, in part because it introduces fewer hyperparameters and allows for flexible pruning rates across the network [24]. However, recent works [33, 14, 34] have identified a key failure mode, *layer-collapse*, for existing pruning algorithms using global-masking. Layer-collapse occurs when an algorithm prunes all parameters in a single weight layer even when prunable parameters remain elsewhere in the network. This renders the network untrainable, evident by sudden drops in the achievable accuracy for the network as shown in Fig. 1. To gain insight into the phenomenon of layer-collapse we will define some useful terms inspired by a recent paper studying the failure mode [34].

Given a network, *compression ratio* ($\rho$) is the number of parameters in the original network divided by the number of parameters remaining after pruning. For example, when the compression ratio $\rho = 10^3$, then only one out of a thousand of the parameters remain after pruning. *Max compression* ($\rho_{\max}$) is the maximal possible compression ratio for a network that doesn't lead to layer-collapse. For example, for a network with $L$ layers and $N$ parameters, $\rho_{\max} = N/L$, which is the compression ratio associated with pruning all but one parameter per layer. *Critical compression* ($\rho_{\mathrm{cr}}$) is the maximal compression ratio a given algorithm can achieve without inducing layer-collapse. In particular, the critical compression of an algorithm is always upper bounded by the max compression of the network: $\rho_{\mathrm{cr}} \leq \rho_{\max}$. This inequality motivates the following axiom we postulate any successful pruning algorithm should satisfy.

**Axiom.** *Maximal Critical Compression. The critical compression of a pruning algorithm applied to a network should always equal the max compression of that network.*

In other words, this axiom implies a pruning algorithm should never prune a set of parameters that results in layer-collapse if there exists another set of the same cardinality that will keep the network trainable. To the best of our

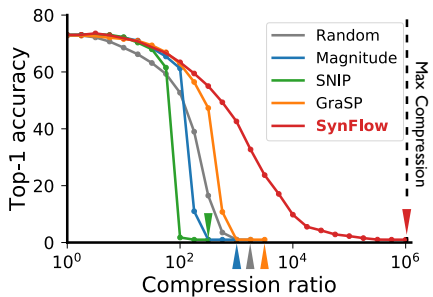

Figure 1: **Layer-collapse leads to a sudden drop in accuracy.** Top-1 test accuracy as a function of the compression ratio for a VGG-16 model pruned at initialization and trained on CIFAR-100. Colored arrows represent the critical compression of the corresponding pruning algorithm. Only our algorithm, SynFlow, reaches the theoretical limit of max compression (black dashed line) without collapsing the network. See Sec. 7 for more details on the experiments.

knowledge, no existing pruning algorithm with global-masking satisfies this simple axiom. Of course any pruning algorithm could be modified to satisfy the axiom by introducing specialized layer-wise pruning rates. However, to retain the benefits of global-masking [24], we will formulate an algorithm, Iterative Synaptic Flow Pruning (SynFlow), which satisfies this property by construction. SynFlow is a natural extension of magnitude pruning, that preserves the total flow of synaptic strengths from input to output rather than the individual synaptic strengths themselves. We will demonstrate that not only does the SynFlow algorithm achieve Maximal Critical Compression, but it consistently matches or outperforms existing state-of-the-art pruning algorithms (as shown in Fig. 1 and in Sec. 7), all while not using the data.

Throughout this work, we benchmark our algorithm, SynFlow, against two simple baselines, random scoring and scoring based on weight magnitudes, as well as two state-of-the-art single-shot pruning algorithms, Single-shot Network Pruning based on Connection Sensitivity (SNIP) [13] and Gradient Signal Preservation (GraSP) [14]. SNIP [13] is a pioneering algorithm to prune neural networks at initialization by scoring weights based on the gradients of the training loss. GraSP [14] is a more recent algorithm that aims to preserve gradient flow at initialization by scoring weights based on the Hessian-gradient product. Both SNIP and GraSP have been thoroughly benchmarked by [14] against other state-of-the-art pruning algorithms that involve training [2, 35, 10, 11, 36, 21, 20], demonstrating competitive performance.

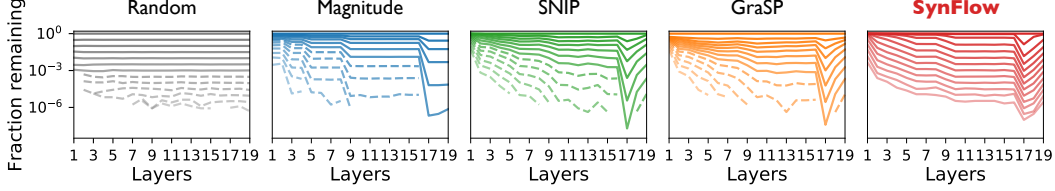

Figure 2: **Where does layer-collapse occur?** Fraction of parameters remaining at each layer of a VGG-19 model pruned at initialization with ImageNet over a range of compression ratios ($10^n$ for $n = 0, 0.5, \ldots, 6.0$). A higher transparency represents a higher compression ratio. A dashed line indicates that there is at least one layer with no parameters, implying layer-collapse has occurred.

## 4 Conservation laws of synaptic saliency

In this section, we will further verify that layer-collapse is a key obstacle to effective pruning at initialization and explore what is causing this failure mode. As shown in Fig. 2, with increasing compression ratios, existing random, magnitude, and gradient-based pruning algorithms will prematurely prune an entire layer making the network untrainable. Understanding why certain score metrics lead to layer-collapse is essential to improve the design of pruning algorithms.

Random pruning prunes every layer in a network by the same amount, evident by the horizontal lines in Fig. 2. With random pruning the *smallest layer*, the layer with the least parameters, is the first to be fully pruned. Conversely, magnitude pruning prunes layers at different rates, evident by the staircase pattern in Fig. 2. Magnitude pruning effectively prunes parameters based on the variance of their initialization, which for common network initializations, such as Xavier [37] or Kaiming [38], are inversely proportional to the width of a layer [34]. With magnitude pruning the *widest layers*, the layers with largest input or output dimensions, are the first to be fully pruned. Gradient-based pruning algorithms SNIP [13] and GraSP [14] also prune layers at different rates, but it is less clear what the root cause for this preference is. In particular, both SNIP and GraSP aggressively prune the *largest layer*, the layer with the most trainable parameters, evident by the sharp peaks in Fig. 2. Based on this observation, we hypothesize that gradient-based scores averaged within a layer are inversely proportional to the layer size. We examine this hypothesis by constructing a theoretical framework grounded in flow networks. We first define a general class of gradient-based scores, prove a conservation law for these scores, and then use this law to prove that our hypothesis of inverse proportionality between layer size and average layer score holds exactly.

**A general class of gradient-based scores.** *Synaptic saliency* is a class of score metrics that can be expressed as the Hadamard product

$$\mathcal{S}(\theta) = \frac{\partial \mathcal{R}}{\partial \theta} \odot \theta, \tag{1}$$

where $\mathcal{R}$ is a scalar loss function of the output $y$ of a feed-forward network parameterized by $\theta$. When $\mathcal{R}$ is the training loss $\mathcal{L}$, the resulting synaptic saliency metric is equivalent (modulo sign) to $-\frac{\partial \mathcal{L}}{\partial \theta} \odot \theta$, the score metric used in Skeletonization [1], one of the first network pruning algorithms. The resulting metric is also closely related to $\left| \frac{\partial \mathcal{L}}{\partial \theta} \odot \theta \right|$ the score used in SNIP [13], $- \left( H \frac{\partial \mathcal{L}}{\partial \theta} \right) \odot \theta$ the score used in GraSP, and $\left( \frac{\partial \mathcal{L}}{\partial \theta} \odot \theta \right)^2$ the score used in the pruning after training algorithm Taylor-FO [28]. When $\mathcal{R} = \langle \frac{\partial \mathcal{L}}{\partial y}, y \rangle$, the resulting synaptic saliency metric is closely related to $\text{diag}(H)\theta \odot \theta$, the score used in Optimal Brain Damage [2]. This general class of score metrics, while not encompassing, exposes key properties of gradient-based scores used for pruning.

**The conservation of synaptic saliency.** All synaptic saliency metrics respect two surprising conservation laws, which we prove in Appendix 9, that hold at any initialization and step in training.

**Theorem 1.** *Neuron-wise Conservation of Synaptic Saliency. For a feedforward neural network with continuous, homogeneous activation functions, $\phi(x) = \phi'(x)x$, (e.g. ReLU, Leaky ReLU, linear), the sum of the synaptic saliency for the incoming parameters (including the bias) to a hidden neuron ($\mathcal{S}^{in} = \langle \frac{\partial \mathcal{R}}{\partial \theta^{in}}, \theta^{in} \rangle$) is equal to the sum of the synaptic saliency for the outgoing parameters from the hidden neuron ($\mathcal{S}^{out} = \langle \frac{\partial \mathcal{R}}{\partial \theta^{out}}, \theta^{out} \rangle$).*

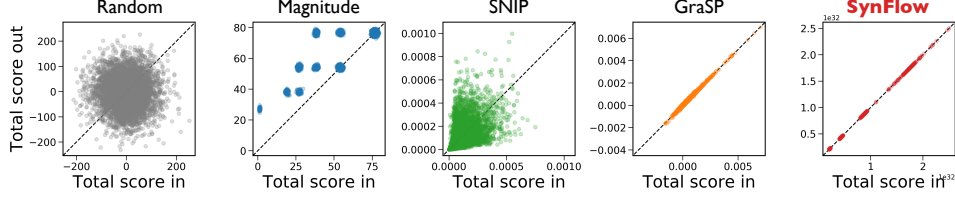

Figure 3: **Neuron-wise conservation of score.** Each dot represents a hidden unit from the feature-extractor of a VGG-19 model pruned at initialization with ImageNet. The location of each dot corresponds to the total score for the unit's incoming and outgoing parameters, $(\mathcal{S}^{\text{in}}, \mathcal{S}^{\text{out}})$. The black dotted line represents exact neuron-wise conservation of score.

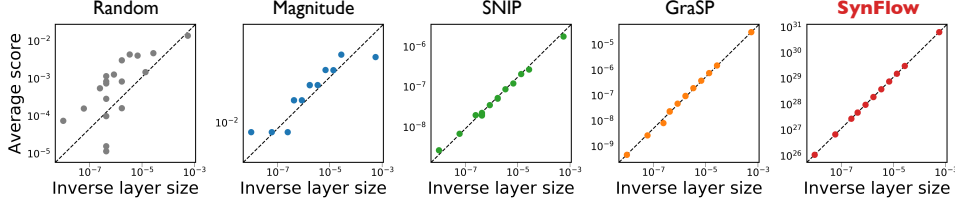

Figure 4: **Inverse relationship between layer size and average layer score.** Each dot represents a layer from a VGG-19 model pruned at initialization with ImageNet. The location of each dot corresponds to the layer's average score[4] and inverse number of elements. The black dotted line represents a perfect linear relationship.

**Theorem 2.** *Network-wise Conservation of Synaptic Saliency. The sum of the synaptic saliency across any set of parameters that exactly[3] separates the input neurons $x$ from the output neurons $y$ of a feedforward neural network with homogeneous activation functions equals $\langle \frac{\partial \mathcal{R}}{\partial x}, x \rangle = \langle \frac{\partial \mathcal{R}}{\partial y}, y \rangle$.*

For example, when considering a simple feedforward network with biases, then these conservation laws imply the non-trivial relationship: $\langle \frac{\partial \mathcal{R}}{\partial W^{[l]}}, W^{[l]} \rangle + \sum_{i=l}^{L} \langle \frac{\partial \mathcal{R}}{\partial b^{[i]}}, b^{[i]} \rangle = \langle \frac{\partial \mathcal{R}}{\partial y}, y \rangle$. Similar conservation properties have been noted in the network complexity [39], implicit regularization [40], and network interpretability [41, 42] literature with some highlighting the potential applications to pruning [9, 43]. While the previous literatures have focused on attribution to the input pixels, or have ignored bias parameters, or have only considered the laws at the layer-level, we have formulated neuron-wise conservation laws that are more general and applicable to any parameter, including biases, in a network. Remarkably, these conservation laws of synaptic saliency apply to modern neural network architectures and a wide variety of neural network layers (e.g. dense, convolutional, pooling, residual) as visually demonstrated in Fig. 3. In Appendix 10 we discuss the specific setting of these conservation laws to parameters immediately preceding a batch normalization layer.

**Conservation and single-shot pruning leads to layer-collapse.** The conservation laws of synaptic saliency provide us with the theoretical tools to validate our earlier hypothesis of inverse proportionality between layer size and average layer score as a root cause for layer-collapse of gradient-based pruning methods. Consider the set of parameters in a layer of a simple, fully connected neural network. This set would exactly separate the input neurons from the output neurons. Thus, by the network-wise conservation of synaptic saliency (theorem 2), the total score for this set is constant for all layers, implying the average is inversely proportional to the layer size. We can empirically evaluate this relationship at scale for existing pruning methods by computing the total score for each layer of a model, as shown in Fig. 4. While this inverse relationship is exact for synaptic saliency, other closely related gradient-based scores, such as the scores used in SNIP and GraSP, also respect this relationship. This validates the empirical observation that for a given compression ratio, gradient-based pruning methods will disproportionately prune the largest layers. Thus, if the compression ratio is large enough and the pruning score is only evaluated once, then a gradient-based pruning method will completely prune the largest layer leading to layer-collapse.

# 5 Magnitude pruning avoids layer-collapse with conservation and iteration

Having demonstrated and investigated the cause of layer-collapse in single-shot pruning methods at initialization, we now explore an iterative pruning method that appears to avoid the issue entirely. Iterative Magnitude Pruning (IMP) is a recently proposed pruning algorithm that has proven to be successful in finding extremely sparse trainable neural networks at initialization (winning lottery tickets) [10, 11, 12, 44, 45, 46, 47]. The algorithm follows three simple steps. First train a network, second prune parameters with the smallest magnitude, third reset the unpruned parameters to their initialization and repeat until the desired compression ratio. While simple and powerful, IMP is impractical as it involves training the network several times, essentially defeating the purpose of constructing a sparse initialization. That being said it does not suffer from the same catastrophic layer-collapse that other pruning at initialization methods are susceptible to. Thus, understanding better how IMP avoids layer-collapse might shed light on how to improve pruning at initialization.

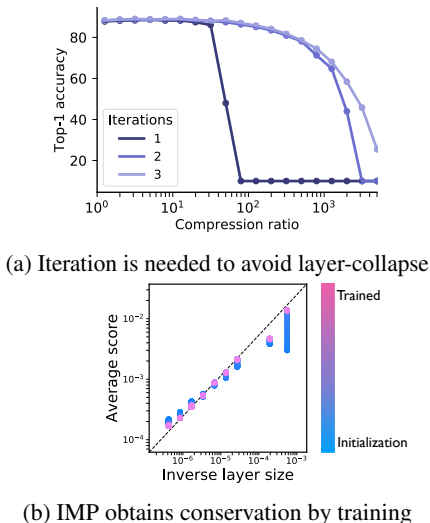

(a) Iteration is needed to avoid layer-collapse

(b) IMP obtains conservation by training

Figure 5: **How IMP avoids layer collapse.** (a) Multiple iterations of training-pruning cycles is needed to prevent IMP from suffering layer-collapse. (b) The average square magnitude scores per layer, originally at initialization (blue), converge through training towards a linear relationship with the inverse layer size after training (pink), suggesting layer-wise conservation. All data is from a VGG-19 model trained on CIFAR-10.

As has been noted previously [10, 11], iteration is essential for stabilizing IMP. In fact, without sufficient pruning iterations, IMP *will* suffer from layer-collapse, evident in the sudden accuracy drops for the darker curves in Fig. 5a. However, the number of pruning iterations alone cannot explain IMP's success at avoiding layer-collapse. Notice that if IMP didn't train the network during each prune cycle, then, no matter the number of pruning iterations, it would be equivalent to single-shot magnitude pruning. Thus, something very critical must happen to the magnitude of the parameters during training, that when coupled with sufficient pruning iterations allows IMP to avoid layer-collapse. We *hypothesize* that gradient descent training effectively encourages the scores to observe an approximate layer-wise conservation law, which when coupled with sufficient pruning iterations allows IMP to avoid layer-collapse.

**Gradient descent encourages conservation.** To better understand the dynamics of the IMP algorithm during training, we will consider a differentiable score $\mathcal{S}(\theta_i) = \frac{1}{2}\theta_i^2$ algorithmically equivalent to the magnitude score. Consider these scores throughout training with gradient descent on a loss function $\mathcal{L}$ using an infinitesimal step size (i.e. gradient flow). In this setting, the temporal derivative of the parameters is equivalent to $\frac{d\theta}{dt} = -\frac{\partial \mathcal{L}}{\partial \theta}$, and thus the temporal derivative of the score is $\frac{d}{dt}\frac{1}{2}\theta_i^2 = \frac{d\theta_i}{dt} \odot \theta_i = -\frac{\partial \mathcal{L}}{\partial \theta_i} \odot \theta_i$. Surprisingly, this is a form of synaptic saliency and thus the neuron-wise and layer-wise conservation laws from Sec. 4 apply. In particular, this implies that for any two layers $l$ and $k$ of a simple, fully connected network, then $\frac{d}{dt}||W^{[l]}||_F^2 = \frac{d}{dt}||W^{[k]}||_F^2$. This invariance has been noticed before by [40] as a form of implicit regularization and used to explain the empirical phenomenon that trained multi-layer models can have similar layer-wise magnitudes. In the context of pruning, this phenomenon implies that gradient descent training, with a small enough learning rate, encourages the squared magnitude scores to converge to an approximate layer-wise conservation, as shown in Fig. 5b.

**Conservation and iterative pruning avoids layer-collapse.** As explained in section 4, conservation alone leads to layer-collapse by assigning parameters in the largest layers with lower scores relative to parameters in smaller layers. However, if conservation is coupled with iterative pruning, then when the largest layer is pruned, becoming smaller, then in subsequent iterations the remaining parameters of this layer will be assigned higher relative scores. With sufficient iterations, conservation coupled with iteration leads to a self-balancing pruning strategy allowing IMP to avoid layer-collapse. This insight on the importance of conservation and iteration applies more broadly to other algorithms with exact or approximate conservation properties. Indeed, concurrent work empirically confirms that iteration improves the performance of SNIP [48, 49].

# 6 A data-agnostic algorithm satisfying Maximal Critical Compression

In the previous section we identified two key ingredients of IMP's ability to avoid layer-collapse: (i) approximate layer-wise *conservation* of the pruning scores, and (ii) the *iterative* re-evaluation of these scores. While these properties allow the IMP algorithm to identify high performing and highly sparse, trainable neural networks, it requires an impractical amount of computation to obtain them. Thus, we aim to construct a more efficient pruning algorithm while still inheriting the key aspects of IMP's success. So what are the essential ingredients for a pruning algorithm to avoid layer-collapse and provably attain Maximal Critical Compression? We prove the following theorem in Appendix 9.

**Theorem 3.** *Iterative, positive, conservative scoring achieves Maximal Critical Compression. If a pruning algorithm, with global-masking, assigns positive scores that respect layer-wise conservation and if the prune size, the total score for the parameters pruned at any iteration, is strictly less than the cut size, the total score for an entire layer, whenever possible, then the algorithm satisfies the Maximal Critical Compression axiom.*

**The Iterative Synaptic Flow Pruning (SynFlow) algorithm.** Theorem 3 directly motivates the design of our novel pruning algorithm, SynFlow, that provably reaches Maximal Critical Compression. First, the necessity for iterative score evaluation discourages algorithms that involve backpropagation on batches of data, and instead motivates the development of an efficient data-independent scoring procedure. Second, positivity and conservation motivates the construction of a loss function that yields positive synaptic saliency scores. We combine these insights to introduce a new loss function (where $\mathbb{1}$ is the all ones vector and $|\theta^{[l]}|$ is the element-wise absolute value of parameters in the $l^{\text{th}}$ layer),

$$\mathcal{R}_{\text{SF}} = \mathbb{1}^T \left( \prod_{l=1}^{L} |\theta^{[l]}| \right) \mathbb{1} \tag{2}$$

that yields the positive, synaptic saliency scores ($\frac{\partial \mathcal{R}_{\text{SF}}}{\partial \theta} \odot \theta$) we term Synaptic Flow. For a simple, fully connected network (i.e. $f(x) = W^{[N]} \ldots W^{[1]} x$), we can factor the Synaptic Flow score for a parameter $w_{ij}^{[l]}$ as

$$\mathcal{S}_{\text{SF}}(w_{ij}^{[l]}) = \left[ \mathbb{1}^\intercal \prod_{k=l+1}^{N} \left| W^{[k]} \right| \right]_i \left| w_{ij}^{[l]} \right| \left[ \prod_{k=1}^{l-1} \left| W^{[k]} \right| \mathbb{1} \right]_j . \tag{3}$$

This perspective demonstrates that Synaptic Flow score is a generalization of magnitude score ($|w_{ij}^{[l]}|$), where the scores consider the product of synaptic strengths flowing through each parameter, taking the inter-layer interactions of parameters into account. In fact, this more generalized magnitude has been discussed previously in literature as a path-norm [50]. The Synaptic Flow loss, equation (2), is the $l_1$-path norm of a network and the synaptic flow score for a parameter is the portion of the norm through the parameter. We use the Synaptic Flow score in the Iterative Synaptic Flow Pruning (SynFlow) algorithm summarized in the pseudocode below.

---

**Algorithm 1:** Iterative Synaptic Flow Pruning (SynFlow).

---

**Input:** network $f(x; \theta_0)$, compression ratio $\rho$, iteration steps $n$

0: $f(x; \theta_0)$ ;                                                           ▷Set model to eval mode[a]
1: $\mu = \mathbb{1}$ ;                                                          ▷Initialize binary mask
**for** $k$ *in* $[1, \ldots, n]$ **do**
   2: $\theta_\mu \leftarrow \mu \odot \theta_0$ ;                                      ▷Mask parameters
   3: $\mathcal{R} \leftarrow \mathbb{1}^T \left( \prod_{l=1}^{L} |\theta_\mu^{[l]}| \right) \mathbb{1}$ ;                       ▷Evaluate SynFlow objective
   4: $\mathcal{S} \leftarrow \frac{\partial \mathcal{R}}{\partial \theta_\mu} \odot \theta_\mu$ ;                                 ▷Compute SynFlow score
   5: $\tau \leftarrow (1 - \rho^{-k/n})$ percentile of $\mathcal{S}$ ;                       ▷Find threshold
   6: $\mu \leftarrow (\tau < \mathcal{S})$ ;                                           ▷Update mask
**end**
7: $f(x; \mu \odot \theta_0)$ ;                                                   ▷Return masked network

---

[a]For pruning at initialization, whether a model is in eval or train mode can have a significant impact on a pruning algorithm's performance, as the batch normalization buffers have not been learned. This is further explained in Appendix 10.

Given a network $f(x; \theta_0)$ and specified compression ratio $\rho$, the SynFlow algorithm requires only one additional hyperparameter, the number of pruning iterations $n$. We demonstrate in Appendix 12, that an exponential pruning schedule ($\rho^{-k/n}$) with $n = 100$ pruning iterations essentially prevents layer-collapse whenever avoidable (Fig. 1), while remaining computationally feasible, even for large networks.

**Computational cost of SynFlow.** The computational cost of a pruning algorithm can be measured by the number of forward/backward passes (#iterations × #examples per iteration). We always run the data-agnostic SynFlow with 100 iterations, implying it takes 100 passes no matter the dataset. SNIP and GraSP each involve one iteration, but use ten times the number of classes per iteration requiring 1000, 2000, and 10,000 passes for CIFAR-100, Tiny-ImageNet, and ImageNet respectively.

## 7 Experiments

We empirically benchmark the performance of our algorithm, SynFlow (red), against the baselines random pruning and magnitude pruning, as well as the state-of-the-art algorithms SNIP [13] and GraSP [14]. In Fig. 6, we test the five algorithms on 12 distinct combinations of modern architectures (VGG-11, VGG-16, ResNet-18, WideResNet-18) and datasets (CIFAR-10, CIFAR-100, Tiny ImageNet) over an exponential sweep of compression ratios ($10^\alpha$ for $\alpha = [0, 0.25, \ldots, 3.75, 4]$). See Appendix 13 for more details and hyperparameters of the experiments. Consistently, SynFlow outperforms the other algorithms in the high compression regime ($10^{1.5} < \rho$) and demonstrates more stability, as indicated by its tight intervals. SynFlow is also quite competitive in the low compression regime ($\rho < 10^{1.5}$). Although SNIP and GraSP can partially outperform SynFlow in this regime, both methods suffer from layer-collapse as indicated by their sharp drops in accuracy.

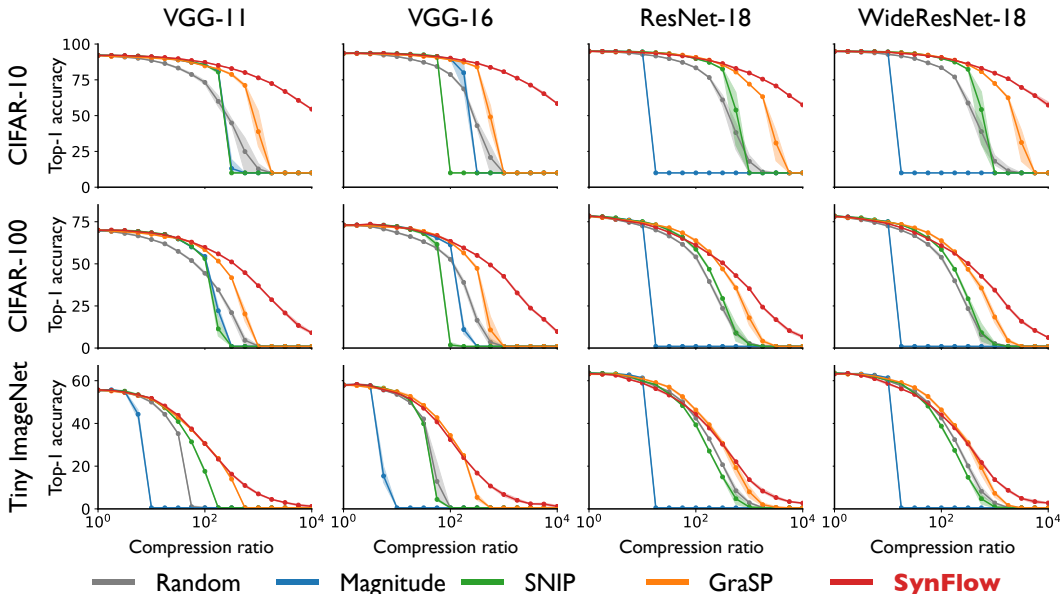

Figure 6: **SynFlow consistently outperforms other pruning methods in high compression regimes avoiding layer collapse.** Top-1 test accuracy as a function of different compression ratios over 12 distinct combinations of models and datasets. We performed three runs with the same hyperparameter conditions and different random seeds. The solid line represents the mean, the shaded region represents area between minimum and maximum performance of the three runs.

**Comparing to expensive iterative pruning algorithms.** Theorem 3 states that iteration is a necessary ingredient for any pruning algorithm, elucidating the success of iterative magnitude pruning and concurrent work on iterative versions of SNIP [48, 49]. As shown in Fig. 7, iteration helps SNIP avoid early layer-collapse, but with a multiplicative computational cost. Additionally, these iterative versions of SNIP still suffer from layer-collapse, long before reaching max compression, which SynFlow is provably guaranteed to reach.

**Ablation studies.** The SynFlow algorithm demonstrates that we do not need to use data to match, and at times outperform, data-dependent pruning methods at initialization such as SNIP and GraSP. This result challenges the effectiveness of how existing algorithms use data at initialization, and provides a concrete algorithmic baseline that any future algorithms that prune at initialization using data should surpass. A recent follow-up work [51] confirms our observation that SynFlow is competitive with SNIP and GraSP even in the low-compression regime and for large-scale datasets (ImageNet) and models (ResNet-50). This work also performs careful ablation studies that offer concrete evidence supporting the theoretical motivation for SynFlow and insightful observations for further improvements of the algorithm. See Appendix 11 for a more detailed discussion on these ablation studies presented in [51].

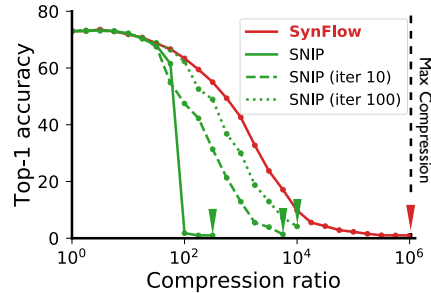

Figure 7: **Iteration improves SNIP, but layer-collapse still occurs.** Top-1 test accuracy as a function of the compression ratio for a VGG-16 model pruned at initialization and trained on CIFAR-100.

## 8 Conclusion

In this paper, we developed a unifying theoretical framework that explains why existing pruning algorithms at initialization suffer from layer-collapse. We applied our framework to elucidate how iterative magnitude pruning [10] overcomes layer-collapse to identify winning lottery tickets at initialization. Building on the theory, we designed a new data-agnostic pruning algorithm, SynFlow, that provably avoids layer-collapse and reaches Maximal Critical Compression. Finally, we empirically confirmed that our SynFlow algorithm consistently matches or outperforms existing algorithms across 12 distinct combinations of models and datasets, despite the fact that our algorithm is data-agnostic and requires no pre-training. Promising future directions for this work are to (i) explore a larger space of potential pruning algorithms that satisfy Maximal Critical Compression, (ii) harness SynFlow as an efficient way to compute appropriate per-layer compression ratios to combine with existing scoring metrics, and (iii) incorporate pruning as a part of neural network initialization schemes. Overall, our data-agnostic pruning algorithm challenges the existing paradigm that data must be used, at initialization, to quantify which synapses of a neural network are important.

## Broader Impact

Neural network pruning has the potential to increase the energy efficiency of neural network models and decrease the environmental impact of their training. It also has the potential to allow for trained neural network models to be more easily deployed on edge devices such as mobile phones. Our work explores neural network pruning mainly from a theoretical angle and thus these impacts are not directly applicable to our work. However, future work might be able to realize these potentials and thus must consider their impacts more carefully.

## Acknowledgments and Disclosure of Funding

We thank Jonathan M. Bloom, Weihua Hu, Javier Sagastuy-Brena, Chaoqi Wang, Guodong Zhang, Chengxu Zhuang, and members of the Stanford Neuroscience and Artificial Intelligence Laboratory for helpful discussions. We thank the Stanford Data Science Scholars program (DK), the Burroughs Wellcome, Simons and James S. McDonnell foundations, and an NSF career award (SG) for support. This work was funded in part by the IBM-Watson AI Lab. D.L.K.Y is supported by the McDonnell Foundation (Understanding Human Cognition Award Grant No. 220020469), the Simons Foundation (Collaboration on the Global Brain Grant No. 543061), the Sloan Foundation (Fellowship FG-2018-10963), the National Science Foundation (RI 1703161 and CAREER Award 1844724), the DARPA Machine Common Sense program, and hardware donation from the NVIDIA Corporation.

## Footnotes

[2]All code is available at `github.com/ganguli-lab/Synaptic-Flow`.

[3]Every element of the set is needed to separate the input neurons from the output neurons.

[4]For GraSP we used the absolute value of the average layer score so that we could plot on a log-log plot.

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
