[Supplementary Material]

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

*Proof.* Consider the $j^{\text{th}}$ hidden neuron of a network with outgoing parameters $\theta_{ij}^{\text{out}}$ and incoming parameters $\theta_{jk}^{\text{in}}$, such that $\frac{\partial \mathcal{R}}{\partial \phi(z_j)} = \sum_i \frac{\partial \mathcal{R}}{\partial z_i} \theta_{ij}^{\text{out}}$ and $z_j = \sum_k \theta_{jk}^{\text{in}} \phi(z_k)$ where there exists a bias parameter $\theta_b^{\text{in}} = b_j$ and a neuron in each layer with the activation $\phi_b = 1$. The sum of the synaptic saliency for the outgoing parameters is

$$\mathcal{S}^{\text{out}} = \sum_i \frac{\partial \mathcal{R}}{\partial \theta_{ij}^{\text{out}}} \theta_{ij}^{\text{out}} = \sum_i \frac{\partial \mathcal{R}}{\partial z_i} \phi(z_j) \theta_{ij}^{\text{out}} = \left( \sum_i \frac{\partial \mathcal{R}}{\partial z_i} \theta_{ij}^{\text{out}} \right) \phi(z_j) = \frac{\partial \mathcal{R}}{\partial \phi(z_j)} \phi(z_j). \quad (4)$$

The sum of the synaptic saliency for the incoming parameters is

$$\mathcal{S}^{\text{in}} = \sum_k \frac{\partial \mathcal{R}}{\partial \theta_{jk}^{\text{in}}} \theta_{jk}^{\text{in}} = \sum_k \frac{\partial \mathcal{R}}{\partial z_j} \phi(z_k) \theta_{jk}^{\text{in}} = \frac{\partial \mathcal{R}}{\partial z_j} \left( \sum_k \theta_{jk}^{\text{in}} \phi(z_k) \right) = \frac{\partial \mathcal{R}}{\partial z_j} z_j. \quad (5)$$

When $\phi$ is homogeneous, then $\frac{\partial \mathcal{R}}{\partial \phi(z_j)} \phi(z_j) = \frac{\partial \mathcal{R}}{\partial z_j} z_j$. $\qquad \square$

**Theorem 2.** *Network-wise Conservation of Synaptic Saliency. The sum of the synaptic saliency across any set of parameters that exactly separates the input neurons $x$ from the output neurons $y$ of a feedforward neural network with homogenous activation functions equals $\langle \frac{\partial \mathcal{R}}{\partial x}, x \rangle = \langle \frac{\partial \mathcal{R}}{\partial y}, y \rangle$.*

*Proof.* We begin by defining the set of neurons ($V$) and the set of prunable parameters ($E$) for a neural network.

Consider a subset of the neurons $S \subset V$, such that all output neurons $y_c \in S$ and all input neurons $x_i \in V \backslash S$. Consider the set of parameters cut by this partition

$$C(S) = \{ \theta_{uv} \in E : u \in S, v \in V \backslash S \}. \quad (6)$$

By theorem 1, we know that that sum of the synaptic saliency over $C(S)$ is equal to the sum of the synaptic saliency over the set of parameters adjacent to $C(S)$ and between neurons in $S$, $\{ \theta_{tu} \in E : t \in S, u \in \partial S \}$. Continuing this argument, then eventually we get that this sum must be equal to the sum of the synaptic saliency over the set of parameters incident to the output neurons $y$, which is

$$\sum_{c,d} \frac{\partial \mathcal{R}}{\partial \theta_{cd}} \theta_{cd} = \sum_{c,d} \frac{\partial \mathcal{R}}{\partial y_c} \phi(z_d) \theta_{cd} = \sum_c \frac{\partial \mathcal{R}}{\partial y_c} \left( \sum_d \theta_{cd} \phi(z_d) \right) = \sum_c \frac{\partial \mathcal{R}}{\partial y_c} y_c = \langle \frac{\partial \mathcal{R}}{\partial y}, y \rangle. \quad (7)$$

We can repeat this argument iterating through the set $V \backslash S$ till we reach the input neurons $x$ to show that this sum is also equal to $\langle \frac{\partial \mathcal{R}}{\partial x}, x \rangle$. $\qquad \square$

**Theorem 3.** *Iterative, positive, conservative scoring achieves Maximal Critical Compression. If a pruning algorithm, with global-masking, assigns positive scores that respect layer-wise conservation and if the prune size, the total score for the parameters pruned at any iteration, is strictly less than the cut size, the total score for an entire layer, whenever possible, then the algorithm satisfies the Maximal Critical Compression axiom.*

*Proof.* We prove this theorem by contradiction. Assume there is an iterative pruning algorithm that uses positive, layer-wise conserved scores and maintains that the prune size at any iteration is less than the cut size whenever possible, but doesn't satisfy the Maximal Critical Compression axiom.

At some iteration the algorithm will prune a set of parameters containing a subset separating the input neurons from the output neurons, despite there existing a set of the same cardinality that does not lead to layer-collapse. By theorem 2, the total score for the separating subset is $\langle \frac{\partial \mathcal{R}}{\partial y}, y \rangle$, which implies by the positivity of the scores, that the total prune size is at least $\langle \frac{\partial \mathcal{R}}{\partial y}, y \rangle$. This contradicts the assumption that the algorithm maintains that the prune size at any iteration is always strictly less than the cut size whenever possible. □

## 10 Pruning with batch normalization

In section 4, we demonstrated how the conservation laws of synaptic saliency lead to an inverse proportionality between layer size and average layer score, $\langle \mathcal{S}_i^{[l]} \rangle = C/N^{[l]}$, for models with homogeneous activation functions. Here, we extend this analysis of layer-collapse to models with batch normalization.

We first review basic mathematical properties of batch normalization. Let $\theta^{\text{in}}$ be the incoming parameters to a neuron with batch normalization, and $x^{(n)}$ be the activation of the preceding layer given the $n^{\text{th}}$ input from a batch $\mathcal{B}$. The output of the batch normalization (before the affine transformation) at this neuron is,

$$\text{BN}(\theta^{\text{in}} x^{(n)}) = \frac{\theta^{\text{in}} x^{(n)} - \mu_{\mathcal{B}}}{\sigma_{\mathcal{B}}},$$

where $\mu_{\mathcal{B}} = \text{E}_{\mathcal{B}}[\theta^{\text{in}} x^{(n)}]$ is the sample mean and $\sigma_{\mathcal{B}}^2 = \text{Var}_{\mathcal{B}}[\theta^{\text{in}} x^{(n)}]$ is the sample variance computed over the batch of data $\mathcal{B}$. Since both mean ($\mu_{\mathcal{B}}$) and variance ($\sigma_{\mathcal{B}}$) scale linearly with respect to the weights, the output of a batch normalization layer is invariant under scaling of incoming parameters: $\text{BN}(\alpha\theta^{\text{in}} x^{(n)}) = \text{BN}(\theta^{\text{in}} x^{(n)})$ for all non-zero $\alpha$. Consequently, any scalar loss function of the output of a network is invariant to scaling these parameters as well, $\mathcal{R}(\alpha\theta^{in}) = \mathcal{R}(\theta^{in})$. This invariance leads to a well known [52, 53] geometric property on the partial gradients of the loss with respect to these parameters: *the sum of Synaptic Saliency for the incoming parameters to a hidden neuron with batch normalization is zero, $\langle \frac{\partial \mathcal{R}}{\partial \theta^{in}}, \theta^{in} \rangle = 0$.*

This geometric property has important implications for synaptic saliency scores in the presence of batch normalization. Crucially, batch normalization breaks the conservation laws for synaptic saliency introduced in section 4 affecting. In fact, batch normalization introduces a new conservation law: the sum of the synaptic saliency scores into a neuron with batch normalization is zero. This property has two important implications: (1) Recall that SynFlow scores are non-negative synaptic saliency, and thus in order to satisfy the previous summation property, all SynFlow scores for parameters preceding a batch normalization layer must be zero. Thus, in order to avoid this failure mode, SynFlow scores are computed in eval mode, which effectively removes batch normalization at initialization. (2) Our previous understanding for the cause of layer-collapse due to SNIP and GraSP was based on an inverse relationship between the layer size and average layer score due to the conservation laws of synaptic saliency. Indeed, as shown in Fig. 8, we confirm that the inverse proportionality still holds empirically even in the presence of batch normalization.

(a) Neuron-wise conservation of score.  (b) Layer size and average layer score.

Figure 8: Here we consider a VGG-19 model with batch normalization pruned at initialization by SNIP and GraSP in train mode using ImageNet. (a) Each dot represents a hidden unit and the location corresponds to the total score for the unit's incoming and outgoing parameters. The black dotted line represents exact neuron-wise conservation of score. (b) Each dot represents a layer and the location corresponds to the layer's average score and inverse number of elements. The black dotted line represents a perfect linear relationship.

## 11 Ablation studies

Our SynFlow algorithm does not make use of any data, yet it matches and at times outperforms other pruning algorithms at initialization that do, notably SNIP and GraSP whose design were motivated by the preservation of the loss [13] and gradient norm [14] respectively. Thus, the success of SynFlow, without a data requirement, motivates the question of what exact elements of these pruning algorithms at initialization matter? A recent work following up on SynFlow [51] has performed ablation studies of pruning at initialization algorithms, identifying the empirical facts that: (1) there is of course still a gap to close to match the accuracy-sparsity tradeoff of much more computationally complex pruning "after" training algorithms that make repeated use of the data and the resultant learned weights; (2) the performance of pruning at initialization methods are unaffected by re-initialization of the weights or within layer shuffling of the mask.

While these ablation studies are insightful, their interpretation that the results "undermine the claimed justifications" [51] of the SynFlow algorithm is incorrect. In fact, one of their ablation studies directly supports the theoretical motivation for the SynFlow algorithm, i.e. to specifically avoid layer-collapse, which it proveably does. When they inverted the SynFlow score so as to prune the most important, rather than most unimportant connections first, the performance of the resulting pruned models dropped immediately for SynFlow. In contrast, the inversion of the corresponding scores results in a relatively moderate performance drop for SNIP, and an unnoticeable drop for GraSP. SynFlow's performance sensitivity to score inversion arises because pruning the *most* important connections identified by the SynFlow score specifically *encourages* layer-collapse instead of avoiding it, yielding a direct decrement in the performance of the pruned models. Thus this ablation study provides direct evidence that SynFlow scores do indeed quantify the relative importance of parameters *across* different layers, for the purpose of avoiding layer collapse.

In two different ablation studies the authors of [51] notice that re-initializing the weights or within layer shuffling of the mask has minimal impact on the performance of convolutional models pruned by SynFlow in low-compression regimes (though we note that unpublished work from the authors shows the impact of shuffling appears to be more significant for fully connected networks). While this observation is interesting, it again does not undermine the claimed theoretical justification of SynFlow, which is, simply the understanding of when pruning algorithms lead to layer collapse, and the development of an algorithm that provably avoids layer collapse. Importantly, since we are performing global pruning across all layers, obtaining the correct per-layer sparsity given a model and a desired compression ratio is a non-trivial task in and of itself. To the best of our knowledge, neither their empirical ablation study nor existing algorithms provide a concrete way to provably achieve maximal critical compression. At this point in time, SynFlow provides, to our knowledge, the only known method of global pruning for achieving that by providing scores whose relative importance *across* layers yields per layer sparsity levels that provably avoid layer collapse at any compression ratio, including at maximal critical compression. Thus, SynFlow remains to be the state-of-the-art algorithm in high-compression regimes and our theoretical framework about layer-collapse provides a solid foundation for guiding the principled design of future algorithms.

Overall our combined theory and empirics, and the ablation studies, also raise intriguing questions about: (1) if and how data might be used more effectively at initialization to prune; (2) whether alternate methods might be able to compute effective per-layer sparsity levels for global pruning at initialization, while still maintaining a good accuracy sparsity tradeoff, especially at high compression ratios; (3) the elucidation of theoretical principles other than the avoidance of layer collapse that could lead to improved global pruning at initialization without data, especially at low compression ratios; (4) the development of global pruning at initialization algorithms that yield more optimal choices of which specific synapses to prune *within* a layer, after effective *across* layer sparsity levels have been determined, either with SynFlow or any other method.

## 12 Hyperparameters choices for the SynFlow algorithm

Motivated by Theorem 3, we can now choose a practical, yet effective, number of pruning iteration ($n$) and schedule for the compression ratios ($\rho_k$) applied at each iteration ($k$) for the SynFlow algorithm. Two natural candidates for a compression schedule would be either linear ($\rho_k = \frac{k}{n}\rho$) or exponential ($\rho_k = \rho^{\frac{k}{n}}$). Empirically we find that the SynFlow algorithm with 100 pruning iterations and an

exponential compression schedule satisfies the conditions of theorem 3 over a reasonable range of compression ratios ($10^n$ for $0 \leq n \leq 3$), as shown in Fig. 9b. This is not true if we use a linear schedule for the compression ratios, as shown in Fig. 9a. Interestingly, Iterative Magnitude Pruning also uses an exponential compression schedule, but does not provide a thorough explanation for this hyperparameter choice [10].

(a) Linear compression schedule           (b) Exponential compression schedule

Figure 9: **Choosing the number of pruning iterations and compression schedule for SynFlow.** Maximum ratio of prune size with cut size for increasing number of pruning iterations for SynFlow with a linear (left) or exponential (right) compression schedule. Higher transparency represents higher compression ratios. The black dotted line represents the maximal prune size ratio that can be obtained while still satisfying the conditions of theorem **??**. All data is from a VGG-19 model at initialization using ImageNet.

**Potential numerical instability.** The SynFlow algorithm involves computing the SynFlow objective, $\mathcal{R}_{\text{SF}} = \mathbb{1}^T \left( \prod_{l=1}^{L} |\theta^{[l]}| \right) \mathbb{1}$, whose singular values may vanish or explode exponentially with depth $L$. This may lead to potential numerical instability for very deep networks, although we did not observe this for the models presented in this paper. One way to address this potential challenge would be to appropriately scale network parameters at each layer to maintain stability. Because the SynFlow algorithm is scale invariant at each layer $\theta^{[l]}$, this modification will not effect the performance of the algorithm. An alternative approach, implemented by Frankle et al. [51], is to increase the precision from single- to double-precision floating points when computing the scores.

## 13 Experimental details

An open source version of our code and the data used to generate all the figures in this paper are available at `github.com/ganguli-lab/Synaptic-Flow`.

### 13.1 Pruning algorithms

All pruning algorithms we considered in our experiments use the following two steps: (i) scoring parameters, and (ii) masking parameters globally across the network with the lowest scores. Here we describe details of how we computed scores used in each of the pruning algorithms.

**Random:** We sampled independently from a standard Gaussian.

**Magnitude:** We computed the absolute value of the parameters.

**SNIP:** We computed the score $\left| \frac{\partial \mathcal{L}}{\partial \theta} \odot \theta \right|$ using a random subset of the training dataset with a size ten times the number of classes, namely 100 for CIFAR-10, 1000 for CIFAR-100, 2000 for Tiny ImageNet, and 10000 for ImageNet. The score was computed in train mode on a batch of size 256 for CIFAR-10/100, 64 for Tiny ImageNet, and 16 for ImageNet, then summed across batches to obtain the score used for pruning.

**GraSP:** We computed the score $\left(H\frac{\partial\mathcal{L}}{\partial\theta}\right)\odot\theta$ using a random subset of the training dataset with a size ten times the number of classes, namely 100 for CIFAR-10, 1000 for CIFAR-100, 2000 for Tiny ImageNet, and 10000 for ImageNet. The score was computed in train mode on a batch of size 256 for CIFAR-10/100, 64 for Tiny ImageNet, and 16 for ImageNet, then summed across batches to obtain the score used for pruning.

**SynFlow:** We applied the pseudocode 1 with 100 pruning iterations motivated by the theoretical and empirical results discussed in Sec 12.

### 13.1.1 The importance of pruning in train mode for SNIP and GraSP

In an earlier version of this work we computed the score for all pruning algorithms in eval mode. While, the details for whether to score in train or eval mode were not discussed directly in either SNIP [13] or GraSP [14], their original code used train mode. We found that for both these algorithms this difference is actually an important detail, especially for larger datasets. Both SNIP and GraSP involve computing data-dependent gradients during scoring, however, because we score at initialization, the batch normalization buffers are independent of the data and thus ineffective at stabilizing the gradients.s

### 13.2 Model architectures

We adapted standard implementations of VGG-11 and VGG-16 from OpenLTH, and ResNet-18 and WideResNet-18 from PyTorch models. We considered all weights from convolutional and linear layers of these models as prunable parameters, but did not prune biases nor the parameters involved in batch normalization layers. For convolutional and linear layers, the weights were initialized with a Kaiming normal strategy and biases to be zero.

### 13.3 Training hyperparameters

Here we provide hyperparameters that we used to train the models presented in Fig. 1 and Fig. 6. These hyperparameters were chosen for the performance of the original model and were not optimized for the performance of the pruned networks.

| | VGG-11 | | VGG-16 | | ResNet-18 | | WideResNet-18 | |
|---|---|---|---|---|---|---|---|---|
| | CIFAR-10/100 | Tiny ImageNet | CIFAR-10/100 | Tiny ImageNet | CIFAR-10/100 | Tiny ImageNet | CIFAR-10/100 | Tiny ImageNet |
| Optimizer | momentum | momentum | momentum | momentum | momentum | momentum | momentum | momentum |
| Training Epochs | 160 | 100 | 160 | 100 | 160 | 100 | 160 | 100 |
| Batch Size | 128 | 128 | 128 | 128 | 128 | 128 | 128 | 128 |
| Learning Rate | 0.1 | 0.01 | 0.1 | 0.01 | 0.01 | 0.01 | 0.01 | 0.01 |
| Learning Rate Drops | 60, 120 | 30, 60, 80 | 60, 120 | 30, 60, 80 | 60, 120 | 30, 60, 80 | 60, 120 | 30, 60, 80 |
| Drop Factor | 0.1 | 0.1 | 0.1 | 0.1 | 0.2 | 0.1 | 0.2 | 0.1 |
| Weight Decay | $10^{-4}$ | $10^{-4}$ | $10^{-4}$ | $10^{-4}$ | $5\times10^{-4}$ | $10^{-4}$ | $5\times10^{-4}$ | $10^{-4}$ |

The original codebase had an issue in the generator used to pass parameters to the optimizer, which effectively multiplied the learning rate by three. This error was used in the training of all models.