[Reviews · NeurIPS 2020]

Review 1

Summary and Contributions: This work propose a novel pruning criteria called synaptic saliency which is based on Hadamard product between the weight magnitudes and the gradients. The concept is extended to a data agnostic pruning algorithm and an iterative pruning algorithm.

Strengths: The method is very well-motivated with sound insights of preventing catastrophic layer-collapse. The authors also connects the idea to explain the success of magnitude pruning, which is also interesting observation. The authors prove the superiority of SynFlow theoretically and empirically. Given the completeness of the paper, I believe it's a work that should be accepted.

Weaknesses: It only provides comparison on random pruning algorithms in the iterate pruning experiments, while I believe it's acceptable given the scope of this paper.

Correctness: Yes.

Clarity: Yes.

Relation to Prior Work: Yes.

Reproducibility: Yes

Additional Feedback: ========== Post-rebuttal ========== Thank you for the response. I have less confidence in the present experiments now due to two reasons: 1. I agree with R3 that the bug in the baseline models (SNIP and GraSP) is quite serious and it would be better that the authors fix it for all the experiments, as the current response only provides part of them. 2. As R2 and R3 mentioned, the iterative process could be the major reason of the effectiveness of preventing layer collapse, which is also shown in Figure B in the response. For sure that the iterative process could not solve layer collapse entirely, while it would be better that the authors provide comparison to stronger baselines and ablation study. I like the contributions of this paper including identification of layer collapse, the interesting ideas in building the SynFlow algorithm, explanation about the success of IMP, and the plausible improvements given the current experiments. However, given the above reasons I would decrease my score to 6 unfortunately.


Review 2

Summary and Contributions: The paper investigates unstructured pruning in neural networks at initialization. It shows that most approaches suffer from layer collapse where a full layer is pruned, leading to catastrophic performance. Papers shows that gradient based pruning approaches respect the conservation laws: their saliency score is conserved at every hidden units and layers in a neural net. It then argues that layer collapse is a result of the conservation laws that apply to gradient based pruning approaches and one-shot pruning. Authors then empirically demonstrates that that iterative magnitude pruning avoids layer-collapse. Retraining encourages the magnitude scores to observe a conservation law, which combined with iterative pruning leads to increase of the saliency scores for the largest layer. Finally, it proposes SynFlow, a data-independent gradient based pruning approach. SynFlow. SynFlow proposes a score function which is data-independent that respects the convervation laws and an iterative estimation procedure of the saliency to avoid layer-collapse. Authors then validates SynFlow on the CIFAR10/100/Tiny-ImageNet datasets. Empircal evaluation shows that SynFlow outperforms previous pruning at initialization approaches such as SNIP or GraSP.

Strengths: The paper identify layer collapse as a clear bottleneck in pruning approaches a propose an algorithm to address this issue. The empirical evaluation validates the proposed approaches. Both the identification of layer collapse in approaches that prune at initialization and the SynFlow algorithm could be of interest to community. The reason of the iterative magnitude pruning (IMP) effectiveness is also an open question. Authors propose a plausible explanation for IMP good performances and provide empirical evidences supporting their claims.

Weaknesses: The main limitation of this work is the absence of comparison with lottery ticket approaches. In particular, the authors claim in the abstract that they ‘identify highly sparse trainable subnetworks at initialization, without ever training’. However, it is unclear if the sparse network found by SynFlow leads to similar performance than the subbnetwork identified through lottery ticket procedure. SynFlow requires to re-estimate the saliency scores while previous approaches such as SNIP and GraSP perform a one-shot pruning. It would be informative to discuss the computation cost associated with the different approaches. How would SynFlow compare in term of performance SNIP or GraSP using iterative estimation of their saliency scores?

Correctness: Claims and method appear correct to me.

Clarity: Paper is clear and pleasant to read.

Relation to Prior Work: Related work is clearly discussed.

Reproducibility: Yes

Additional Feedback: In figure 3, why do the conservation laws only seems to hold for GraSP and SynFlow and not for SNIP? Is it because of the absolute value used when computing the saliency criterion? In figure 5, is the performance reported after pruning or after retraining of the pruned network? --------------- Post rebuttal update Thanks for your rebuttal which addresses most of my concerns. I will keep my original and positive rating.


Review 3

Summary and Contributions: This paper proposes a new criteria for pruning neural networks at initialization, and even without ever accessing the data. The authors first define the Maximal Critical Compression, i.e., the maximal compression rate before resulting in layer-collapse for a given pruning algorithm. Then the authors propose synaptic saliency, a class of gradient-based scores (previous works SNIP and GraSP also fall in this class), which is conserved at every hidden unit and layer of neural network. The conservation law reveals that single-shot pruning will result in layer-collapse. As a mitigation, the authors propose Synflow, which iteratively prune the network at initialization without access to any data. The experiments are conducted on CIFAR10, CIFAR-100, and Tiny-ImageNet, showing that Synflow outperforms the baseline by a big margin.

Strengths: - The authors propose a general class of gradient-based scores for pruning neural networks at initialization, which unifies the previous works SNIP and GraSP in this class. - The proposed method is well-motivated based on the observation of layer-collapse and conservation law. - The experiments are thorough and carefully designed. - The study on Iterative Magnitude Pruning is interesting and offers insights on why IMP performs so well in practice.

Weaknesses: Method: - Theorem 1 is not new, and similar results were already presented in previous work [Liang et al., 2019]. - Theorem 1 does not hold for neural networks with bias terms. For the proof of Theorem 1, In L171, there are no bias terms that appeared in computing z_j. However, for ResNet and VGGNet they all have bias terms at each layer, so this theorem does not apply for them. - I do not think data-independent is a good feature in designing pruning algorithms since the network architecture should depend on the data distribution. For example, when the input data is very sparse and most of its dimensions are not correlated with the prediction, then you can probably prune a lot at the input layer. Therefore, I am not convinced with the motivation of data-independent pruning. Experiments: - For magnitude pruning, do you apply it on a pretrained network or a randomly initialized network? I am a bit surprised that it performs almost the worst among these methods. - The comparisons between SNIP, GraSP, and SynFlow is not fair, because SynFlow prunes the network iteratively. So, I would suggest the authors include comparisons with the iterative version of SNIP. - Why GraSP performs much worse than SNIP? When computing the saliency of GraSP and SNIP, it seems that you did not set the model to training mode. Is this a bug? The results will be more convincing if you can adopt the same settings as used in the baseline papers. Though this is an interesting paper, I am inclined to reject this paper at the current stage, as I am not convinced with the empirical results due to the potential bugs in the implementation of SNIP and GraSP. Besides, Theorem 1 does not apply to the networks studied in the experiments, as they all have bias terms. The authors should clarify this difference, and further experiments may be needed to study the role of bias terms computing the saliency score. Liang, Tengyuan, et al. "Fisher-rao metric, geometry, and complexity of neural networks." The 22nd International Conference on Artificial Intelligence and Statistics. 2019. ==================== After Rebuttal ====================== Thanks to the authors' response. Unfortunately, I am still inclining to rejection due to the following reasons: 1. [Theorem 1] Although the authors' response has addressed my concerns about applying Thm 1 on networks with bias terms, it is still not straightforward to see if it applies to BatchNorm layers. A detailed proof will be necessary. Besides, Thm 1 is obvious given the results in [Liang et al., 2019]. 2. [Bug is critical] More importantly, the results of all the baseline models (SNIP and GraSP) were wrong due to the bugs in the code. Though the authors claimed they've rerun all of the experiments during the rebuttal period, only part of the results are provided in the rebuttal due to the one-page limitation. In general, I believe it's fairly unfair to accept a paper with buggy implementations of all the baseline models, let alone all the baseline models have public code available. I believe this is an interesting paper, but due to the aforementioned reasons, I vote for rejection at the current stage. I strongly recommend the authors resubmit this work to a near-future conference with more convincing results and detailed proofs.

Correctness: The method itself is correct and empirical methodology is also correct. The proofs of the Theorems are correct, but Theorem 1 does not apply to the networks (VGGNets and ResNets) adopted in the experiments.

Clarity: The paper is overall well written, but the structure can be improved. E.g., Section 5 can be moved to appendix, as it is not very coherent with the paragraphs before and after it. The authors should also highlight their methods more.

Relation to Prior Work: Yes.

Reproducibility: Yes

Additional Feedback:


Review 4

Summary and Contributions: This paper developed a unifying theoretical framework that explains why existing single-shot pruning algorithms at initialization suffer from layer-collapse. It also designed a new data-agnostic pruning algorithm, SynFlow, that provably avoids layer-collapse and reaches Maximal Critical Compression. Lastly, it achives remarkable and consistent performance gain on several benchmark models and datasets.

Strengths: 1. This work has a theoretical framework and solid experiments to support its arguments. 2. it achives remarkable and consistent performance gain on several benchmark models and datasets when the compression ratio is very high.

Weaknesses: 1. When the compression ratio is lower than 10, SynFlow's performance is comparable to other SOTA methods. One thing I am confused about is that do we really need such a high compress ratio, e.g. 100 or higher? Such a high compress ratio hurts the model's performance significantly and makes its performance not quite meaningful.

Correctness: Yes.

Clarity: Good writing overall but it can be better.

Relation to Prior Work: Yes.

Reproducibility: Yes

Additional Feedback:

[Author Response · NeurIPS 2020]

We thank all reviewers for their careful reviews and positive comments, including: (**R1**) "the method is very well-
motivated with sound insights", (**R2**) "both the identification of layer collapse . . . and the SynFlow algorithm could be
of interest to community", (**R3**) "the study on Iterative Magnitude Pruning is interesting and offers insights", (**R4**) "this
work has a theoretical framework and solid experiments to support its arguments". We now address reviewer concerns:

**Theorem 1 generalizes prior conservation laws.** As we mention on L180/245, restricted versions of our conservation
laws have been noted in the interpretability [39] and implicit regularization [45] literature. We will also cite [Liang et al.]
as suggested by **R3**. Our theorem 1 generalizes these prior laws in three significant ways: (1) We do not limit ourselves
to only gradients of activations [39] or the training loss [45], but consider any "scalar function of the output" L157.
(2) Rather than proving conservation by layer only [Liang et al.], we prove conservation at the stronger neuron-level
and generalize to any cut separating the input from the output (theorem 2). (3) We consider all incoming and outgoing
parameters (weights and biases) allowing us to avoid the assumption that biases are zero ([45], [Liang et al.]) and
understand how conservation applies to a "variety of neural network layers (e.g. dense, convolutional, batchnorm,
pooling, residual)" L186. Most importantly, we are the first to connect conservation laws to network pruning and
elucidate their significance in explaining a multitude of phenomena and in constructing a new pruning algorithm.

**Theorem 1 holds even with biases, batch normalization, and residual connections.** In Theorem 1, we consider $\theta^{\text{in}}$
to encompass *all* incoming parameters including the biases. We can understand **R3**'s confusion, so to clarify our proof,
we will make the notation $z_j = \sum_k \theta_{jk}^{\text{in}} \phi(z_k)$ more explicit by designating a bias parameter $\theta_b^{in} = b_j$ and a neuron in
each layer with the activation $\phi_b = 1$. We will further explain, mathematically and graphically, how our conservation
laws generalize across modern architectures and at any point in training. For example, when considering a simple
feedforward network with biases, then we get the non-trivial relationship: $\langle \frac{\partial \mathcal{L}}{\partial W^{[l]}}, W^{[l]} \rangle + \sum_{i=l}^{L} \langle \frac{\partial \mathcal{L}}{\partial b^{[i]}}, b^{[i]} \rangle = \langle \frac{\partial \mathcal{L}}{\partial y}, y \rangle$.
Nonetheless, standard initialization schemes set biases to zero, thus the simpler version of our conservation law suffices
to analyze the ResNet/VGGNet architectures we consider empirically at initialization, resolving **R3**'s concern.

**Pruning in train mode and an implementation discrepancy in GraSP.** We
thank **R3** for noticing that the original implementations of SNIP and GraSP
prune in train mode. To match the implementation exactly, we updated our code
base and re-ran the results for both algorithms in figures 1 and 6. However the
empirical conclusions with respect to SynFlow have not changed, as noted in
the updated version of figure 1 on the right (Grasp - Fig. A, SNIP - Fig. B) We
further communicated with the authors of GraSP to eliminate any implementation
discrepancies in our GraSP submission code (orange line) and now our updated
implementation (blue line) matches within error of the official baseline (dashed
black line). As **R3** expected, GraSP now better aligns with SNIP (especially with
Tiny-ImageNet where we had first reported poor GraSP performance). However,
the assumption that GraSP should always align with SNIP is incorrect. A recent
preprint [de Jorge, et al. 2020] independently reports that GraSP can perform
much worse than SNIP at high compression ratios (SNIP: $51.3\%$, GraSP: $0.1\%$
at $98\%$ sparsity on VGG19/ImageNet) and the GraSP paper compared to SNIP
at only 3 compression ratios. We sweep over 26 compression ratios representing
one of the most thorough benchmarks of pruning algorithms at initialization.

Fig. A: Agreement with the baseline (VGG16/CIFAR-100: cf. Fig1)

Fig. B: Iterative SNIP (VGG16/CIFAR-100: cf. Fig1)

**Iterative magnitude, SNIP, GraSP pruning.** Our theorem 3 states that iteration is a necessary ingredient for any
pruning algorithm, elucidating the success of iterative magnitude pruning, as **R2** noted, and concurrent work on iterative
versions of SNIP [Verdenius, et al. 2020], [de Jorge, et al. 2020]. We are currently comparing SynFlow to these much
more computationally expensive methods. From initial experiments, iterative SNIP avoids layer collapse, but sacrifices
its performance in low compression regimes underperforming SynFlow (Fig. B).

**Computational cost of SynFlow.** The computational cost of a pruning algorithm can be measured by the number of
forward/backward passes (#iterations × #examples per iteration). We always run the data-agnostic SynFlow with 100
iterations, implying it takes 100 passes no matter the dataset. SNIP and GraSP each involve 1 iteration, but use 10
times the number of classes per iteration requiring 1000, 2000, and 10,000 passes for CIFAR-100, Tiny-ImageNet, and
ImageNet respectively. Iterative versions of them will be multiplicatively more expensive. For example, iterative SNIP
with 100 iterations on CIFAR-100 would require 100,000 passes, whereas SynFlow would only require 100!

**SynFlow's data-agnostic property brings the fields of network pruning and initialization together.** As noted by
**R4**, SynFlow demonstrates the most impressive empirical improvement to other methods in the high compression
regimes. However, even in the low compression regimes SynFlow does on par with other methods without even looking
at the data, which we believe to be a major accomplishment. This striking capability which we have demonstrated
theoretically and empirically in our work opens up a new direction of sparse initialization via network pruning.

[Meta-Review · NeurIPS 2020]

This paper introduces synaptic saliency for deep network pruning which does not require any dataset. It provides interesting insights. on layer collapse. While there is overall appreciation of the results theoretically there are some concerns about the experiments. The authors have addressed some of the issues and contingent on their addressing the concerns one can recommend acceptance.